# Linear Representation Meta-Reinforcement Learning for Instant Adaptation

## Abstract

This paper introduces *Fast Linearized Adaptive Policy* (FLAP), a new meta-reinforcement learning (meta-RL) method that is able to extrapolate well to out-of-distribution tasks without the need to reuse data from training, and adapt almost instantaneously with the need of only a few samples during testing. FLAP builds upon the idea of learning a shared linear representation of the policy so that when adapting to a new task, it suffices to predict a set of linear weights. A separate adapter network is trained simultaneously with the policy such that during adaptation, we can directly use the adapter network to predict these linear weights instead of updating a meta-policy via gradient descent, such as in prior meta-RL methods like MAML, to obtain the new policy. The application of the separate feed-forward network not only speeds up the adaptation run-time significantly, but also generalizes extremely well to very different tasks that prior Meta-RL methods fail to generalize to. Experiments on standard continuous-control meta-RL benchmarks show FLAP presenting significantly stronger performance on out-of-distribution tasks with up to double the average return and up to 8X faster adaptation run-time speeds when compared to prior methods.

## 1 Introduction

Deep Reinforcement Learning (DRL) has led to recent advancements that allow autonomous agents to solve complex tasks in a wide range of fields (Schulman et al. (2015), Lillicrap et al. (2015a), Levine et al. (2015)). However, traditional approaches in DRL learn a separate policy for each unique task, requiring large amounts of samples. Meta-Reinforcement learning (meta-RL) algorithms provide a solution by teaching agents to implicitly learn a shared structure among a batch of training tasks so that the policy for unseen similar tasks can quickly be acquired (Finn et al. (2017)). Recent progress in meta-RL has shown efforts being made in improving the sample complexity of meta-RL algorithms (Rakelly et al. (2019), Rothfuss et al. (2018)), along with the out-of-distribution performance of meta-RL algorithms during adaptation (Fakoor et al. (2019); Mendonca et al. (2020)). However, most of the existing meta-RL algorithms prioritize sample efficiency at the sacrifice of computational complexity in adaptation, making them infeasible to adapt to fast-changing environments in real-world applications such as robotics.

In this paper, we present **Fast Linearized Adaptive Policy** (FLAP), an off-policy meta-RL method with great generalization ability and fast adaptation speeds. FLAP is built on the assumption that similar tasks share a common linear (or low-dimensional) structure in the representation of the agent's policy, which is usually parameterized by a neural network. During training, we learn the shared linear structure among different tasks using an actor-critic algorithm. A separate *adapter* net is also trained as a supervised learning problem to learn the weights of the output layer for each unique train task given by the environment interactions from the agent. Then when adapting to a new task, we fix the learned linear representation (shared model layers) and predict the weights for the new task using the trained adapter network. An illustration of our approach is highlighted in Figure 2. We highlight our main contributions below:

- **State of the Art Performance.** We propose an algorithm based on learning and predicting the shared linear structures within policies, which gives the strongest results among the meta-RL algorithms and the fastest adaptation speeds. FLAP is the state of the art in all these areas including performance, run-time, and memory usage. As is shown in Fig-

ure 1, the FLAP algorithm outperforms most of the existing meta-RL algorithms including MAML (Finn et al. (2017)), PEARL (Rakelly et al. (2019)) and MIER (Mendonca et al. (2020)) in terms of both adaptation speed and average return. Further results from our experiments show that FLAP acquires adapted policies that perform much better on out-of-distribution tasks at a rapid run-time adaptation rate up to 8X faster than prior methods.

- **Prediction rather than optimization.** We showcase a successful use of prediction via adapter network rather than optimization with gradient steps (Finn et al. (2017)) or the use of context encoders (Rakelly et al. (2019)) during adaptation. This ensures that different tasks would have policies that are different from each other, which boosts the out-of-distribution performance, while gradient-based and context-based methods tend to produce similar policies for all new tasks. Furthermore, the adapter network learns an efficient way of exploration such that during adaptation, only a few samples are needed to acquire the new policy. To our knowledge, this is the first meta-RL method that directly learns and predicts a (linearly) shared structure successfully in adapting to new tasks.

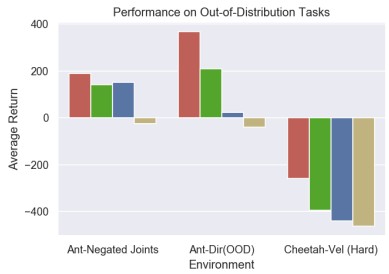
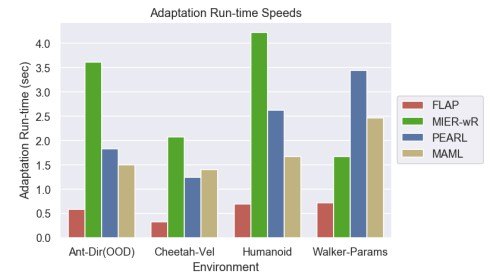

Figure 1: **Strong Experimental Results:** We showcase the performance of meta-RL methods on tasks that are very different from the training tasks to assess the generalization ability of methods. We also analyze the adaptation run-time speed of these methods on tasks that are similar (in-distribution) and tasks that are not very similar (out-of-distribution) to further evaluate these models. Flap presents significantly stronger results compared to prior meta-RL methods.

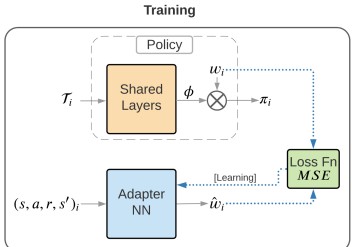
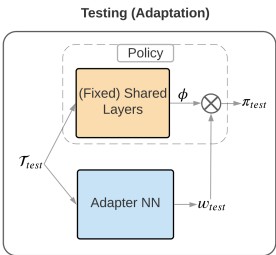

Figure 2: **Overview of our approach:** In training, for different tasks $\{\mathcal{T}_i\}$, we parametrize their policy as $\pi_i = \phi \cdot w_i$, where $\phi \in \mathbb{R}^d$ is the shared linear representation we hope to acquire. In testing (adaptation), we fix the acquired linear representation $\phi$ and directly alter the weights $w_{test}$ by using the output of the feed-forward adapter network.

## 1.1 RELATED WORK

The idea behind learning a shared linear structure is motivated by the great progress of representation learning and function approximation in domains such as robotics (Kober et al. (2013)) and natural language processing (Keneshloo et al. (2018)). Representation learning has seen success in cases of transfer learning RL in different tasks, and multi-agent reinforcement learning (MARL) where different agents may share common rewards or dynamics (Taylor & Stone (2009), Silva et al. (2018)). The idea of learning shared information (embeddings) across different tasks has been investigated deeply in transfer learning, including for example, universal value function approximators (Schaul

et al. (2015)), successor features (Barreto et al. (2017), Hunt et al. (2019)), and invariant feature spaces (Gupta et al. (2017))).

Using a prediction model during meta-testing presents a new idea into meta-RL. The current trend in meta-reinforcement learning has been to either use an encoder based method through the use of context captured in recurrent models (Fakoor et al. (2019)) or variational inference (Rakelly et al. (2019)), or use a variation of model-agnostic meta-learning (MAML) (Finn et al. (2017)), which aims at finding some common parametrization of the policy such that it suffices to take one gradient step in adaption to acquire a good policy. In contrast, FLAP aims at predicting the weights for the policy network directly in adaptation without the need of taking gradient steps or using context and comparing with existing data. This gives better performance in (a) faster adaptation speeds due to the minimal amount of computation required; (b) smaller sample complexity during adaptation due to the better exploration strategy provided by adapter network; (c) strong out-of-distribution generalization ability resulting from the fact that the policy for each new task can be completely different in the last layer (in contrast, the new policies produced by gradient-based algorithms like MAML only differ in a single gradient update, which makes them highly similar, hurting the generalization ability). We believe the idea of predicting instead of fine-tuning via gradient descent can shed light to broader questions in meta-learning.

## 2 BACKGROUND

We consider the standard Markov Decision Process (MDP) reinforcement learning setting defined by a tuple $\mathcal{T} = (\mathcal{S}, \mathcal{A}, \mu, \mu_0, r, T)$, where $\mathcal{S}$ is the state space, $\mathcal{A}$ is the action space, $\mu(s_{t+1}|s_t, a_t)$ is the unknown transition distribution with $\mu_0$ being the initial state distribution, $r(s_t, a_t)$ is the reward function, and $T$ is the episode length. Given a task $\mathcal{T}$, an agent's goal is to learn a policy $\pi : \mathcal{S} \rightarrow \mathcal{A}$ to maximize the expected (discounted) return:

$$q_\gamma(\pi, \mathcal{T}) = \mathbb{E}[\sum_{t=0}^{T-1} \gamma^t r(s_t, a_t)], s_0 \sim \mu_0, a_t \sim \pi(s_t), s_{t+1} \sim \mu(s_t, a_t) \tag{1}$$

We now formalize the meta-RL problem. We are given a set of training tasks $\mathcal{T}_{train} = \{\mathcal{T}_i\}_{i=1}^n$ and test tasks $\mathcal{T}_{test} = \{\mathcal{T}_j\}_{j=1}^m$. In meta-training, we are allowed to sample in total $k$ trajectories $\mathcal{D}_{train}^k$ from the tasks in $\mathcal{T}_{train}$ and obtain some information $\phi(\mathcal{D}_{train}^k)$ that could be utilized for adaptation. The choice of $\phi(\mathcal{D}_{train}^k)$ varies for different algorithms. In meta-testing (or adaptation), we are allowed to sample $l$ trajectories $\mathcal{D}_{test}^l$ from each $\mathcal{T}_i \in \mathcal{T}_{test}$. We would like to obtain a policy $\pi(\phi(\mathcal{D}_{train}^k), \mathcal{D}_{test}^l)$ for each testing task $\mathcal{T}_i$, such that the average expected return for all testing tasks is maximized:

$$\frac{1}{m} \sum_{\mathcal{T}_i \in \mathcal{T}_{test}} q_\gamma(\pi(\phi(\mathcal{D}_{train}^k), \mathcal{D}_{test}^l), \mathcal{T}_i). \tag{2}$$

Here the policy only depends on the training data through the information $\phi(\mathcal{D}_{train}^k)$, and can be different for different test tasks. We are interested in getting the expected return in equation 2 as large as possible with the smallest sample complexity $k, l$. Furthermore, we hope that the information $\phi(\mathcal{D}_{train}^k)$ is reasonably small in size such that the memory needed for testing is also controlled.

We also state that in-distribution tasks typically refer to the test tasks being drawn from the same distribution as the training distribution of tasks whereas out-of-distribution tasks represent the test distribution as very different from the training distribution (in our case, all *out-of-distribution tasks* are completely disjoint from the training set).

## 3 FAST LINEARIZED ADAPTIVE POLICY (FLAP)

In this section, we describe the FLAP algorithm. We first assert and justify our assumption behind the goal of the FLAP algorithm (Section 3.1), and then discuss the high-level flow of the algorithm (Section 3.2).

### 3.1 Assumption and Motivation

Our FLAP algorithm is based on the following assumption on the shared structure between different tasks during training and testing:

**Assumption 3.1** *We assume that there exists some function $\phi : \mathcal{S} \rightarrow \mathbb{R}^{d_1}$ such that for any task in the meta-RL problem $\mathcal{T}_i \in \{\mathcal{T}_{\text{train}} \cup \mathcal{T}_{\text{test}}\}$, there exists a policy $\pi_i$ of the following form that maximizes the expected return $q_\gamma(\pi, \mathcal{T}_i)$:*

$$\pi_i = \sigma(\langle \phi, w_i \rangle + b_i). \tag{3}$$

*Here $w_i \in \mathbb{R}^{d_1 \times d_2}$, $b_i \in \mathbb{R}^{d_2}$. $d_1$ is the latent space dimension and $d_2$ is the dimension of the action space. The function $\phi$ represents a shared structure among all tasks, and $w_i$ and $b_i$ are the unique weight and bias for task $\mathcal{T}_i$. The function $\sigma$ is the activation function depending on the choice of the action space.*

Similar assumptions that constrain the policy space also appear in literature, e.g. Levine et al. (2016); Lillicrap et al. (2015b); Schmidhuber (2019); Mou et al. (2020) and references therein. One can also extend the linear representation in equation 3 to quadratic or higher-order representations for more representational ability and looser asumptions. We compare the performance between linear structures and non-linear structures in Section C.1.3.

### 3.2 Algorithm overview

Given assumption 3.1, the overall workflow of the algorithm is quite straightforward: in meta-training, we would like to learn the shared feature $\phi$ from the training tasks $\mathcal{T}_{train}$; in meta-testing, we utilize the learned linear feature $\phi$ to derive the parameters $w_i, b_i$ for each new task. To speed up the adaptation, we also train a separate feed-forward adapter network simultaneously that takes the trajectories of each task as input and predicts the weights that are learned during training. The full pseudo-code for both meta-training and meta-testing is provided in Appendix A.

#### 3.2.1 Meta-training

In order to learn the linear representation $\phi$ from Assumption 3.1, we use a deep neural network for the policy, sharing all but the output layer among all the train tasks in the training set. Each train task will have a unique output layer during meta-training, and the shared neural network is the feature $\phi$ we hope to learn during training.

Mathematically, we parametrize the policy via a neural network as $\pi(\phi, w_i, b_i) = \sigma(\langle \phi, w_i \rangle + b_i)$, where $w_i$ and $b_i$ are the weights and bias unique to task $i$, and $\sigma$ represents the linear activation for the output layer since for all the tasks we considered in this paper, the action space takes values in the range of $[-\infty, +\infty]$.

FLAP uses the general multi-task objective to train the policy over all the $n$ training tasks during meta-training. We maximize the average returns over all training tasks by learning the policy parameters represented by $\{\phi, \{w_i, b_i\}_{i=1}^n\}$. We would like to find the meta-learned parameters via solving the following optimization problem:

$$\{\phi, \{w_i, b_i\}_{i=1}^n\} = \underset{\{\phi, \{w_i, b_i\}_{i=1}^n\}}{\arg\max} \frac{1}{n} \sum_{i=1}^n q_\gamma(\pi(\phi, w_i, b_i), \mathcal{T}_i). \tag{4}$$

Here $\phi$ is parameterized by a neural network, and $\pi_i$ is parameterized by the parameters $\{\phi, \{w_i, b_i\}_{i=1}^n\}$.

**Actor-Critic.** For solving the optimization problem in equation 4, we apply the off-policy soft actor-critic (SAC) (Haarnoja et al. (2018)) for sample efficiency.

In the actor-critic model, besides the policy network, we also have a critic network which learns a $Q$ function for evaluating the current policy. We linearize the critic network in a similar fashion to the policy by fixing a set of shared layers in the critic network. We show the shared layer structure for both policy and critic network in Figure 3. Linearizing the critic network, though not explicitly

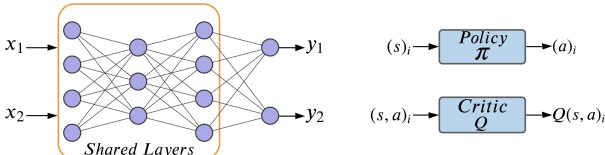

Figure 3: Each task shares a feature mapping in the critic and a different shared feature mapping in the policy network. In this illustration, for the training task 1 input data $x_1$, only the output $y_1$ from its corresponding output layer will be used in the meta-loss function. Similarly for task 2 input data $x_2$, only the output $y_2$ from its corresponding output layer will be used in the meta-loss. This is extended up to $n$ points where $n$ is the number of training tasks used.

required, does not incur any loss in performance from experimental results, and drastically reduces the number of parameters needed to be learned during training for the critic portion of the actor-critic method. We discuss in more detail about the use of the soft actor-critic in Appendix D.1.

**Adapter network.** Learning the shared structure is a standard approach in multi-task transfer learning (Pan & Yang (2009); Asawa et al. (2017); D'Eramo et al. (2020)). The key contribution and distinction of our algorithm is the use of an adapter network for rapid adaptation. We train a separate adapter network simultaneously with the shared structure during meta-training. We define the **SARS** tuple as a tuple containing information from one time step taken on the environment by an agent given a single task, including the current state, action taken, reward received from taking the action given the current state, and the next state the agent enters $(s_t, a_t, s_{t+1}, r(s_t, a_t))$. The adapter net is fed a single tuple or a sequence of the most recent SARS tuples from a task and predicts the unique weights $(\hat{w}_i)$ needed to acquire the new policy for that task. We train the adapter network with mean squared error between the output of the adapter net $(\hat{w}_i)$ obtained from the input SARS tuples, and the set of weights obtained from meta-training contained in each output layer. The process of training the adapter net is shown in Figure 4.

Training the Adapter Network

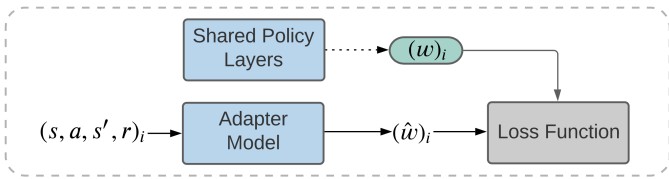

Figure 4: Illustration of training the adapter network. The weights of the corresponding output layer for the training task are flattened and concatenated to form a 1-D target for the output of the adapter model.

### 3.2.2 META-TESTING

In meta-testing, the FLAP algorithm adapts the meta-learned policy parameters $w_i, b_i$ to a new task ($\mathcal{T}_i \in \mathcal{T}_{test}$) with very little data sampled from the new task. Following Assumption 3.1, we assume that the shared structure $\phi$ is already learned well during meta-training. Thus in meta-testing, we fix the structure $\phi$ obtained in training, and only adjust the weights for each test task. Mathematically, we run the following optimization problem to obtain the policy for each new test task $\mathcal{T}_{new} \in \mathcal{T}_{test}$: The objective for the new task ($\mathcal{T}_{new}$) is to find the set of weights that with the shared structure $\phi$ form the adapted policy

$$w_{new}, b_{new} = \arg\max_{w,b} q_\gamma(\pi(\phi, w, b), \mathcal{T}_{new}). \tag{5}$$

We then output $\pi(\phi, w_{new}, b_{new})$ as the policy for the new task.

In our practical implementation and results, we show our contribution that requiring an objective function to be optimized during meta-testing does not necessarily have to be the only approach

Figure 5: The first step taken is arbitrary. Each step following the first is fed into the feed-back loop where the SARS pair is inputted to the adapter network providing a new set of weights for the output layer of the policy.

for meta-RL, though FLAP can still have the optimization in Equation 5 easily integrated in meta-testing. FLAP performs rapid online adaptation by using the adapter network to predict the weights of the output layer as opposed to performing the optimization step. The first time-step taken by the agent is based off the most recent weights learned from the adapter network. This then initiates a feed-back loop where now each time-step gives the adapter network a single tuple or a sequence of the most recent **SARS** tuples which inputted to the adapter network, immediately produces a set of predicted weights to use with the linear feature mapping ($\phi$) to acquire an adapted policy ($\pi_{\text{adapt}} = \phi^T \cdot w_i + b_i$). Only a fixed number of online adaptation steps are needed to acquire the fully adapted policy. Once the last time-step of weights is predicted, FLAP keeps *only* the last set of predictive weights for the rest of the episode. The meta-testing loop is shown in Figure 5.

## 4    EXPERIMENTS

This section presents the main experients of our method. We evaluate FLAP in Section 4.1 by comparing it to prior meta-learning approaches on in-distribution tasks. We then examine how the FLAP algorithm is able to generalize to very different test tasks using out-of-distribution tasks in Section 4.2. We take a holistic view of FLAP as a meta-RL method and analyze its run-time and memory usage on an autonomous agent during the meta-testing stage in Section 4.3. We leave the ablation studies and sparse reward experiments to Appendix C. Finally, we provide thorough discussions on the empirical performance in Section 4.4.

### 4.1    SAMPLE EFFICIENCY AND PERFORMANCE COMPARED TO PRIOR METHODS

We evaluate FLAP on standard meta-RL benchmarks that use in-distribution test tasks (training and testing distributions are shared). These continuous control environments are simulated via the MuJoCo simulator (Todorov et al. (2012)) and OpenAI Gym (Brockman et al. (2016)). Rakelly et al. (2019) constructed a fixed set of meta-training tasks ($\mathcal{T}_{train}$) and a validation set of tasks ($\mathcal{T}_{test}$). These tasks have different rewards and randomized system dynamics (Walker-2D-Params). To enable comparison with these past methods, we follow the evaluation code published by Rakelly et al. (2019) and follow the exact same evaluation protocol. In all the tasks we consider, the adapter network is fed with a single SARS tuple during training and adaptation. Other hyper-paramters of our method are given in Appendix D.3 and we report the result of FLAP on each environment from 3 random seeds. We compare against standard baseline algorithms of MAML (Finn et al. (2017)), RL2 (Duan et al. (2016b)), ProMP (Rothfuss et al. (2018)), PEARL (Rakelly et al. (2019)), MQL (Fakoor et al. (2019)), and MIER (and MIER-wR) (Mendonca et al. (2020)). We obtained the training curves for the first four algorithms from Rakelly et al. (2019) and the latter two from Mendonca et al. (2020).

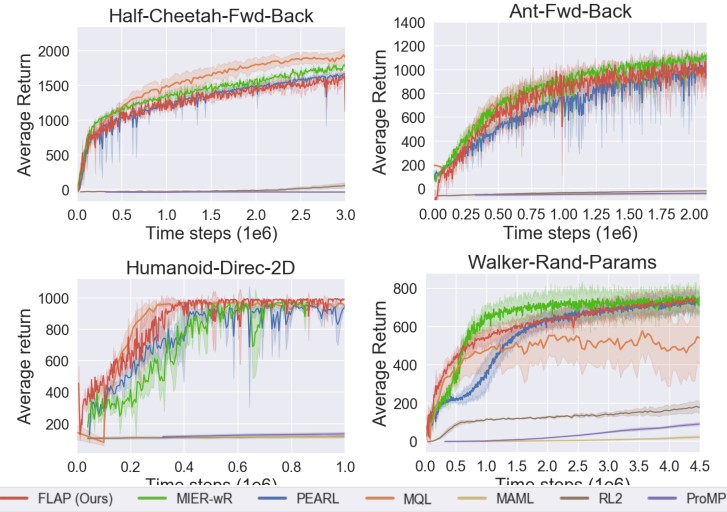

Figure 6: Validation Test-Task performance over the course of the meta-training process on standard in-distribution meta-RL benchmarks. Our algorithm is comparable in all environments to prior meta-RL methods.

## 4.2 GENERALIZATION TO OUT-OF-DISTRIBUTION TASKS

To evaluate generalization, we test on the extrapolated out-of-distribution (OOD) environments, where the train and test distributions are highly dissimilar and completely disjoint, created by Mendonca et al. (2020) and Fakoor et al. (2019), closely following their evaluation protocols. The environments include changes in rewards and dynamics. We run FLAP on 3 random seeds to evaluate the performance. In all the tasks we consider, the adapter network is fed with a single **SARS** tuple during training and adaptation. Hyperparameters are listed in Appendix D.3.

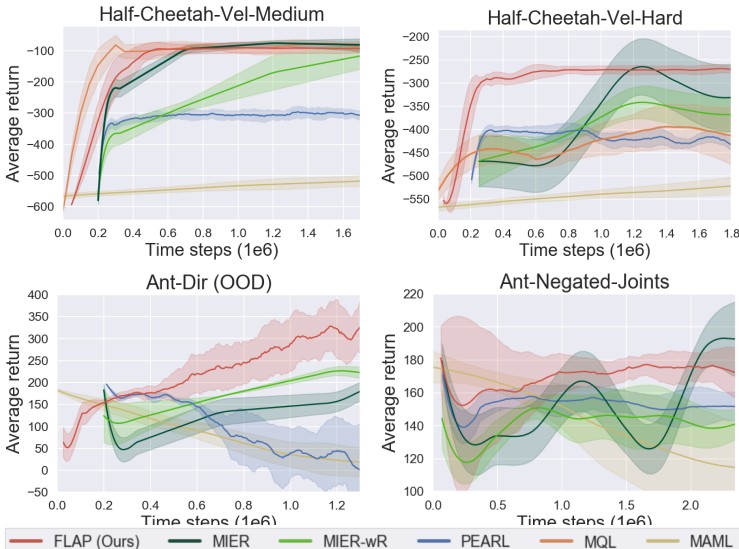

Figure 7: Validation Test-Task performance over the course of the meta-training process on **out-of-distribution** meta-RL benchmarks. Our algorithm is comparable if not better to prior methods in all environments except for the Ant-Negated task.

### 4.3 OVERALL MODEL ANALYSIS

In this section, we assess the run-time of FLAP. We compare our method to MIER-wr and PEARL in meta-testing speeds on benchmarks from both in-distribution and out-of-distribution tasks. MIER and MQL both reuse the training buffer during testing, causing adaptation speeds and memory usage that are far worse than prior methods (MIER took 500+ seconds for meta-testing). We only compare with the most competitive algorithms speed and memory-wise to showcase FLAP's strength. We defer the analysis of storage efficiency to Appendix (B).

To calculate the run-time speeds, we use the Python Gtimer package and a NVIDIA GEFORCE GTX 1080 GPU and time each of the Meta-RL methods on four MuJoCo environments (Ant, Cheetah, Humanoid, Walker) on 30 validation tasks after allowing each algorithm to achieve convergence during meta-training.

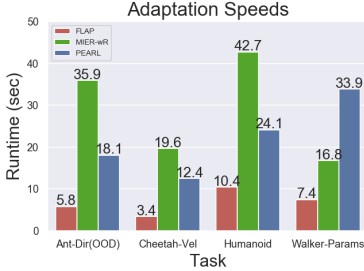

Figure 8: Our method clearly outperforms all other meta-RL algorithms in runtime during adaptation to new tasks.

### 4.4 DISCUSSION

Beyond the experiments on in-distribution and out-of-distribution tasks, we did more experiments and analysis to justify the superiority of FLAP, including ablation studies and extra experiments on MDPs with sparse rewards. We briefly describe the results and take-aways in Section 4.4.1 and 4.4.2, and leave the details on experiments to Appendix C. Lastly, we analyze all the experimental results and summarize the reason for the validity and generalization ability of the adapter network in Section 4.4.3

### 4.4.1 ABLATION STUDIES

We validate the effectiveness of FLAP via a large amount of ablation studies. In particular, we compare the performance under the following different settings:

1. **With adapter net vs Without adapter net:** One may wonder whether the adapter network really boosts the performance compared to the approach of directly learning the linear weights via soft actor-critic during adaptation without the help of the adapter network. As is shown in Appendix C.1.1, although both algorithms converge to approximately the same average reward in the end, the adapter network drastically decreases the sample complexity during the adaptation process and accelerates the speed (at least 30X faster among all the experiments) compared to the algorithm without the adapter.

2. **Shared linear layer vs Non-linear layers:** One may wonder why we assume the policies share only the very last layer instead of the last two or more layers as embeddings. Indeed, experiments in Appendix C.1.3 show that when sharing two (or more) layers, the policies may have too much degree of freedom to converge to the global minimum. The shared linear layer turns out to be the best assumption that balances the performance with the speed.

3. **Single SARS tuple vs a Sequence of SARS tuple as Input of Adapter Net:** All the experiments above work with the adapter net with a single (the most recent) SARS tuple as input. We show in Appendix C.1.4 that if we let the adapter net take a sequence of

the most recent SARS tuples as input, the average return would not be affected on the tasks we considered above. However, we see a reduction in variance with a slightly slower adaptation speed. Overall, taking a longer sequence of SARS tuples as input tends to **stabilize the adaptation** and **reduce the variance** in the sacrifice of runtime. However, for some specific tasks with sparse rewards and similar SARS tuples, it also helps improve the average return, as is discussed in the next section.

### 4.4.2 EXPERIMENTS ON SPARSE REWARD

We show in Appendix C.2 that for tasks with sparse rewards and similar dynamics (e.g. the navigation tasks), the performance of FLAP may deteriorate a little since the resulting policy highly relies on the chosen transition tuple, which may lead to higher variance. However, it is still comparable to PEARL if we let the adapter net take a sequence of SARS tuples as input. This is due to fact that a sequence of SARS tuples provides more information on the context for the adapter, making the resulting policy less dependent on the single chosen transition tuple.

However, for most of the other tasks we considered above, one single transition tuple suffices to provide enough information for the success of adapter network with small variance and strong generalization ability. We also validate it in Appendix C.3 by showing that the loss for adapter net converges in a reliably fast and stable way.

### 4.4.3 ON THE VALIDITY AND GENERALIZATION ABILITY OF FLAP AND THE ADAPTER NETWORK

To summarize the experimental results above, we see that FLAP guarantees a much smaller sample complexity during adaptation, improved generalization ability and faster adaptation speeds. We provide justification as to why FLAP can achieve all the properties separately.

1. **Small Sample Complexity:** From our ablation study in Appendix C.1.1, it is clear that the small sample complexity during meta-testing comes from the introduction of adapter net. We conjecture that during training, the adapter net learns the structure of optimal policies shared by all the training tasks; and during adaptation, the predicted weights encourage the most efficient exploration over potential structures of optimal policies such that the average return converges in a fast and stable way.

2. **Strong generalization:** From our ablation study in Appendix C.1.1, we see that the algorithms with and without adapter converge to a similar average return. Thus we conclude that the generalization ability is not related to the specific design of adapter net, but comes from the shared linear structure. In traditional gradient-based meta-RL approaches like MAML, the new policies are generated from taking one gradient step from the original weights in training, which makes the new policies very similar to the old ones. In contrast, the new policy in FLAP can have a completely different last layer, which guarantees more flexibility in varying policies, and thus a better generalization to new tasks.

3. **Fast adaptation:** Compared to other meta-RL approaches, FLAP only requires evaluating the output of a fixed adapter net, which provides instant adaptation when the adapter net is small enough.

The successful experiments also justify the validity of Assumption 3.1: for all the tasks we consider, they do have some near-optimal policies with shared embedding layers. We believe this could also provide interesting theoretical directions as well.

## 5 CONCLUSION

In this paper, we presented **FLAP**, a new method for meta-RL with significantly faster adaptation run-time speeds and performance. Our contribution in the FLAP algorithm presents a completely new view into meta reinforcement learning by applying linear function approximations and predicting the adaptation weights directly instead of taking gradient steps or using context encoders. We also believe the idea behind the adapter network during meta-testing can be extended to other meta learning problems outside of reinforcement learning.

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

## A    PSEUDO CODE

Algorithm 1 and Algorithm 2 provide the pseudo-code for meta-training and meta-testing (adaptation to a new task) respectively. In meta-training (1), the replay buffers $\mathcal{B}$ store data for each train task and provide random batches for training the soft actor-critic. For each train task $\mathcal{T}_i$ in meta-training, the actor loss $\mathcal{L}_{actor}$ is the same as $J_\pi(\theta, \mathcal{T})$ which is defined in Equation 8. The critic loss $\mathcal{L}_{critic}$ is the same as that used in the soft actor-critic with the Bellman update:

$$\mathcal{L}_{critic} = \mathbb{E}_{(s,a,r,s') \sim \mathcal{B}^i}[Q_\psi(s,a) - (r + \gamma * Q(s', a'))]^2, \quad a' \sim \pi_\theta(a'|s') \qquad (6)$$

The adapter loss $\mathcal{L}_{adapter}$ is the $MSE$ loss. These three networks are all optimized through gradient descent, with learning rates denoted by $\alpha$ for each respective network. The learning rate for each of the three networks is a hyper-parameter predetermined before training.

After meta-training, a meta-learned policy $\pi_\theta$ and trained adapter $\varphi$ are used for adaptation. Algorithm 2 lays out the adaptation process where the policy $\pi_\theta$ is to be adapted. A fixed, predetermined number (T) of adaptation steps are taken. The process is illustrated in Figure 4.

---

**Algorithm 1 FLAP: Meta-Training**

---

**Require:** Training Tasks Batch $\{\mathcal{T}_i\}_{i=1...N}$ from $\rho(\mathcal{T}_{\text{train}})$
1: Initialize replay buffers $\mathcal{B}^i$ for each train task
2: **while** not done **do**
3:     **for** each $\mathcal{T}_i$ **do**
4:         Gather data from $\pi_\theta$ and add to $\mathcal{B}^i$
5:     **end for**
6:     **for** step in train steps **do**
7:         **for** each train task $\mathcal{T}^i$ **do**
8:             Sample batch of data $b^i \sim \mathcal{B}^i$
9:             $\mathcal{L}^i_{actor} = \mathcal{L}^i_{actor}(b^i)$
10:            $\mathcal{L}^i_{critic} = \mathcal{L}^i_{critic}(b^i)$
11:            weights = Flatten($W_{y_i(\phi)}$)
12:            target = Concat($weights, B_{y_i(\phi)}$)
13:            $\mathcal{L}^i_{adapter} = \mathcal{L}^i_{adapter}((s, a, r, s\prime), target)$
14:        **end for**
15:        $\psi_Q \leftarrow \psi_Q - \alpha_Q \nabla_\psi \sum_i(\mathcal{L}^i_{critic})$
16:        $\theta_\pi \leftarrow \theta_\pi - \alpha_\pi \nabla_\theta \sum_i(\mathcal{L}^i_{actor})$
17:        $\varphi \leftarrow \varphi - \alpha_\varphi \nabla_\varphi \sum_i(\mathcal{L}^i_{adapter})$
18:    **end for**
19: **end while**

---

**Algorithm 2 FLAP: Meta-Testing**

---

**Require:** Test Task $\mathcal{T} \sim \rho(\mathcal{T}_{test})$
1: Load policy model with parameter $\theta$ and fix the output layer, load adapter $\varphi$
2: Initialize output layer weights with any output layer from training
3: **for** $t = 1 ... T$ **do**
4:     $a_t \sim \pi_\theta(a_t|s_t)$
5:     $s_{t+1} \sim \rho(s_{t+1}|s_t, a_t)$
6:     $r_t = r(s_t, a_t)$
7:     (weights, bias) = $\varphi(s_t, a_t, s_{t+1}, r_t)$
8:     Update $\theta$ by setting the output layer parameters as (weights, bias).
9: **end for**

---

## B    MEMORY USAGE

For storage analysis of models, we concern ourselves only with the amount of memory needed during meta-testing as the process of meta-learning is to meta-learn a policy that is both lightweight in storage, fast to adapt, and effective in performance on a new task. We assessed the internal

structure of models based off the number of parameters used in each algorithm for the models that are necessary when adapting to a new task. As already noted in Section 4.3, we only use MIER without experience relabeling (Mendonca et al. (2020)) and PEARL (Rakelly et al. (2019)) in comparison with FLAP. We note that though MIER with experience relabeling is better on the Ant-Negated Task in comparison with FLAP, it utilizes the process of *experience relabeling* where the training replay buffer must be used for meta-testing. MQL (Fakoor et al. (2019)) also requires the use of the training buffer for adapting to out-of-distribution tasks. This leads to a highly negative result in both algorithms since the amount of memory necessary for these models to adapt to highly extrapolated tasks grows as the number of meta-train tasks used during training increases and the complexity of the meta-RL problem increases.

PEARL and MIER-wR (without experience relabeling) do not need to reuse the replay buffer however FLAP performs uniformly better than these methods in all the out-of-distribution baselines. The model parameters are detailed in Table 1 and clearly FLAP is at least comparable if not better in the aspect of memory required for adaptation.

| Algorithm | Policy Size | Model Size |
|-----------|-------------|------------|
| FLAP (OURS) | 300x300x300 | 400x400x400 |
| MIER-wR | 300x300x300 | 200x200x200x200 |
| Pearl | 300x300x300 | 200x200x200 |

Table 1: The model and policy work together in meta-testing. The model is typically an extra network for context encoding (PEARL, MIER-wR) or adaptation (FLAP). FLAP is comparable in memory to the most competitive algorithms memory-wise. We use default settings for all algorithms for the memory analysis.

## C ABALATION STUDIES AND FURTHER EXPERIMENTS

In this section, we showcase some more experiments involving FLAP.

### C.1 EXPERIMENTAL DETAILS ON ABLATION STUDY

We conclude our experimental analysis by showcasing ablation studies on the adapter network. We first provide further justification for the use of the adapter network in the first place and evidence that it works with reasonable decreasing loss while the model learns. We then showcase the effect during adaptation of feeding in different input data into the adapter network. We finally finish by showcasing single-point input compared to a sequence of points concatenated together as input to the adapter network and its result on the effectiveness of the adapter network. Analysis of the loss values of the adapter network during training can be found in Appendix C.3.

### C.1.1 WITH ADAPTER NET VS WITHOUT ADAPTER NET: STRONG PERFORMANCE OF THE ADAPTER NETWORK

To analyze the importance of the adapter network, we compare the adaptation (meta-test) results on the Half-Cheetah environment with the use of feed-forward prediction compared to simply using a meta-testing objective and performing SAC objective function gradient updates to iteratively acquire the new policy. Although both methods are able to eventually obtain a strong performing policy, the adapter network drastically decreases the amount of samples needed for adaptation and provides a clear benefit for the FLAP algorithm, as is shown in Figure C.1.1. We also see a dramatic decrease in run-time (at least 30X among all the experiments) when we have the adapter network in the adaptation procedure.

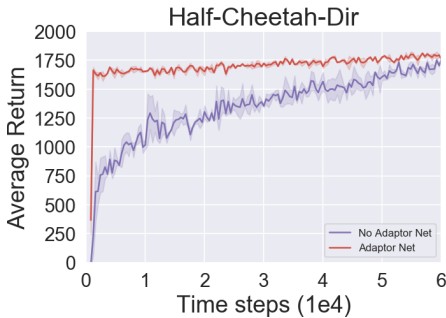

Figure 9: Ablation study for FLAP with vs without adapter network. Although both algorithms converge to similar average return, the algorithm with adapter net converges much faster than the one without adapter.

### C.1.2 SAS INPUT VS SARS INPUT

We test different inputs to the adapter network. We train the adapter using only (state, action, next state) pairs compared to the original complete **SARS** pair to analyze whether the adapter is able to learn the reward function. Given the random and poor performance of the adapter with only $(S, A, S')$ as input, we provide evidence that the adapter net learns valuable information to aid the algorithm during adaptation.

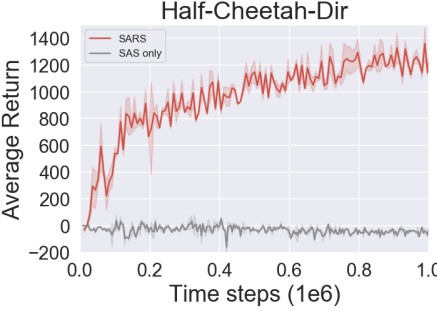

Figure 10: Results of varying the input to the adapter network. In environments such as Half-Cheetah, where tasks differ in the reward function, the reward is a valuable input to the adapter network and helps in its effectiveness for predicting weights.

### C.1.3 NONLINEAR OUTPUT LAYERS

Our idea of maintaining a shared set of layers and a linear output layer can easily be extended beyond the linear case. We motivate this section by showcasing that FLAP does not gain performance from increased expressiveness when extended to the multi-layer output case. We use a two-layer (quadratic) output and reduce the hidden units in each layer from 300, which was used in the linear case, to 50 to keep the total number of output units in both FLAP algorithms approximately consistent. We also use the adapter network (learning non-linear weights during meta-testing from scratch slowed down adaptation speeds) to predict both layer weights. We ran the regular and modified FLAP algorithms on out-of-distribution

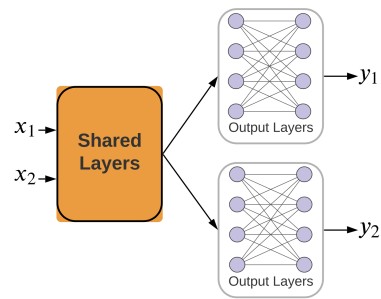

Figure 11: We add an extra layer, with a non-linear activation function, to the original linear output layer to create the non-linear output.

tasks to compare generalization strength of each method. Our implementation of the modified FLAP algorithm is illustrated in Figure 11. Results of the comparison are shown in Figure 12.

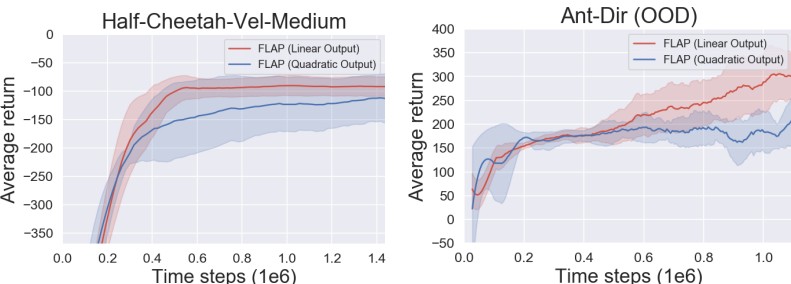

Figure 12: The linear FLAP algorithm outperformed the modified (non-linear) FLAP algorithm.

### C.1.4 SEQUENCE VS. SINGLE POINT OF SARS
TUPLES AS INPUT OF ADAPTER NET

We showcase that using a sequence of tuples (5 tuples) does not improve the performance of the algorithm in our baseline problems and in cases where the linear assumption holds up well, although it offers improvements including reduced variance in less structured cases where the assumption may not hold as strongly such as in sparse reward settings (Appendix C.2). It is important to note that the adapter is batch-wise trained meaning that a random batch of **SARS** tuples are sampled from the replay buffer of the corresponding task and all trained with the same set of target weights. Although the network may be presented a tuple where the state or action is completely new, the adapter is able to generalize and make a strong prediction with just a single point and without sacrifice to performance indicating the self-correcting nature of the adapter network.

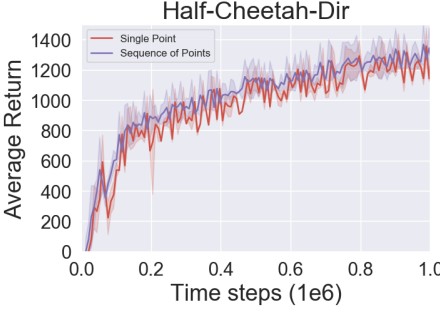

Figure 13: Results of Single Point tuple input vs Sequence of SARS tuples as inputs on the Half-Cheetah environment. The results showcase both methods are comparable in return.

### C.2 SPARSE REWARD ENVIRONMENTS

Sparse Rewards Reinforcement Learning environments require the agent to reason about uncertainty in an environment where different tasks share many common features and seemingly appear identical. We compare against the PEARL algorithm on the same (in-distribution) sparse reward setting (Sparse Point Robot) first introduced by PEARL that requires navigation in a 2D setting to a goal target (Rakelly et al. (2019)).

We mention that although FLAP only needs 1 trajectory of test data, we allow the PEARL algorithm to obtain as many trajectories from the unseen task as necessary to achieve its strongest performance which already requires 20+ trajectories even in such a simple problem. We also run FLAP using a single point adapter input and a sequence of points (5 tuples) as an input to the adapter network. In

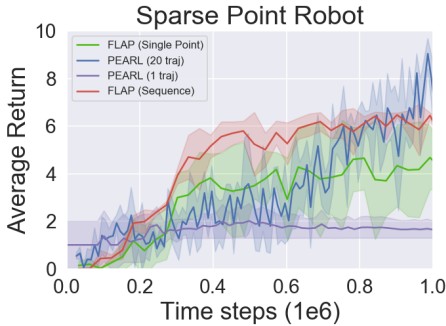

Figure 14: Results of the sparse reward environment. FLAP, with the use of a sequence of inputs to the adapter network, is comparable with PEARL even with the FLAP algorithm using 20x less samples from the test task.

this case, the sequence of points as input helped not only improve the performance but also presented less variance among different runs of the algorithm which indicates that on problems where the linear assumption may not be as strong, such as sparse reward settings, using multiple points as inputs may present improved results. Our algorithm completely outperforms PEARL when both algorithms are only allowed 1 trajectory of data, indicating that FLAP has better scalability especially when more complex sparse reward problems are encountered where more test data is required to be sampled to acquire the new policy.

In conducting the experiment, we followed the PEARL algorithm in training on dense rewards for the sparse point robot and testing on the sparse reward setting to allow for the fair comparison between the algorithms. We state that the success of the FLAP algorithm may be attributed to the fact that the implementation is built on the SAC algorithm which uses entropy to help with exploration, a vital component especially in sparse reward problems.

## C.3 ADAPTER NETWORK LOSS

During training, we captured the loss of the adapter network during training for its ability to train on target train task weights. In the experiments conducted, the loss value typically converged nicely overall with no indication of any high loss values which would have presented conflicting performance on the different baseline tasks. We showcase the loss on the Half-Cheetah environment and the Sparse Point Robot Problem where there are sparse rewards. Although the adapter network trained reasonably on both, it is important to note that when the linear assumption is not as strong, the loss value tends not to converge as smoothly which also leads to variance in the performance of the algorithm on the environment.

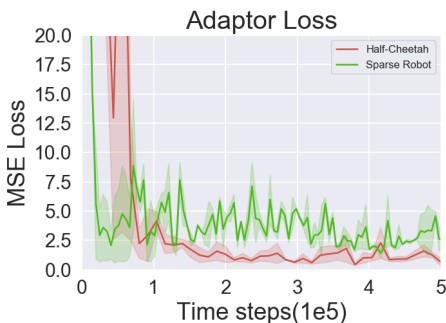

# D  IMPLEMENTATION DETAILS

We provide implementation details including the hyper-parameters used in building FLAP and the settings used for evaluation of algorithms on both in-distribution and out-of-distribution tasks.

## D.1  ACTOR-CRITIC IMPLEMENTATION

We discuss more specifics on the use of the actor-critic algorithm in our model. Specifically, for the actor and critic networks, we parameterize the critic network by $\psi$, leading to $Q_\psi$, and the policy network by $\theta$, leading to $\pi_\theta$, then we can write the objective function for actor-critic algorithms to optimize the policy (actor) network:

$$Q_\psi^{\pi_\theta}(s,\ a,\mathcal{T}) = \frac{1}{N}\sum_{i=1}^{N}\mathbb{E}_{s_t,a_t\sim\pi_\theta,\mathcal{T}_i}[\sum_{t=0}^{T}\gamma^t r(s_{t,\mathcal{T}_i},a_{t,\mathcal{T}_i})|s_{0,\mathcal{T}_i}\ =s,\ a_{0,\mathcal{T}_i}=a] \tag{7}$$

$$J_\pi(\theta,\mathcal{T}) = \frac{1}{N}\sum_{i=1}^{N}[-\mathbb{E}_{s\sim\mathcal{T}_i,a\sim\pi}[Q_\psi^{\pi_\theta}(s,\ a,\mathcal{T})]], \tag{8}$$

where Equation 7 is the Q-value estimation from the critic and the policy loss $J_\pi$ in Equation 8 specifically details the loss function used to optimize the meta-policy parameters. We also employ automatic entropy-tuning for our SAC algorithm to help with exploration and employ two Q-function networks to help reduce over-estimation bias, a technique commonly known as "double-Q-learning" (van Hasselt et al. (2015)).

## D.2  EXPERIMENT DETAILS

The horizon for all environments is 200 time-steps. Evaluation of return is then the average reward of the trajectories where each trajectory is the sum of the rewards collected over its rollout. We use settings from PEARL (Rakelly et al. (2019)) for the in-distribution tasks and the settings from MIER (Mendonca et al. (2020)) for the out-of-distribution tasks. The settings for the environments are listed in the Table. FLAP requires less meta-train tasks for meta-training than prior methods. The number of adaptation steps refers to the number of iterations of the feedback loop illustrated by meta-testing and described in Section 4 before the policy is finally settled. The settings are detailed in Table 2 and Table 3.

Table 2: In-Distribution Settings

| Environment | Meta-train Tasks | Meta-test Tasks | Adaptation Steps |
|---|---|---|---|
| Half-Cheetah-Fwd-Back | 2 | 2 | 30 |
| Ant-Fwd-Back | 2 | 2 | 60 |
| Humanoid-Dir | 5 | 30 | 30 |
| Walker-Rand | 10 | 10 | 60 |

Table 3: Out-of-Distribution Settings

| Environment | Meta-train Tasks | Meta-test Tasks | Adaptation Steps |
|---|---|---|---|
| Cheetah-vel-medium | 15 | 30 | 1 |
| Cheetah-vel-hard | 15 | 30 | 1 |
| Ant-direction | 10 | 30 | 60 |
| Ant-negated-joints | 15 | 15 | 30 |

## D.3  HYPER-PARAMETERS

The hyper-parameters for FLAP are kept mostly fixed across the different tasks except for the number of parameter updates per iteration which is different only for the Ant and Walker environments

(4000). All others use the default of 1000 updates. These hyper-parameters were kept as default values from the PEARL (Rakelly et al. (2019)) soft actor-critic implementation code in the PEARL code base as both FLAP and PEARL are built based off the rllab code base (Duan et al. (2016a)). They are detailed in Table 4.

Table 4: Hyper-parameters

| Parameter | Value |
|---|---|
| Learning Rate | $3 \times 10^{-4}$ |
| Discount Factor | 0.99 |
| Target Update Interval | 1 |
| Target Update Rate | 0.005 |
| Sac Reward Scale | 1.0 |
| Sac Optimizer | Adam |
| Batch-size | 256 |
| Policy Arch. | 300-300-300 |
| Critic Arch | 300-300-300 |
| Adapter Arch | 600-600-600 |
| Adapter Learning Rate | $3 \times 10^{-4}$ |
| Adapter Optimizer | Adam |
| Parameter Updates per Iteration | 1000.4000 |
| Num Points in Sequence Input | 5, 10 |

