# OpenReview forum: "Linear Representation Meta-Reinforcement Learning for Instant Adaptation"
_ICLR.cc/2021/Conference — Reject_

### Official Review · AnonReviewer3 · 2020-10-28
**Simple meta RL approach with competitive performance**

**Rating:** 5
**Confidence:** 3

**Review:**

This paper proposed a meta RL algorithm built upon an assumption that a shared policy with a task-specific final linear layer can maximize expected return for each task.  The proposed method trains the policy using soft actor-critic on training tasks and trains an adapter network to predict task-specific final linear layer from a single timestep transition (s, a, r, s'). The proposed method showed a competitive performance with the previous approaches and showed a faster adaptation speed

Strengths
* This paper showed that a simple approach can achieve a competitive performance on meta reinforcement benchmarks
* The presented method showed faster adaptation compared to previous approaches

Weaknesses
* Motivation for the proposed method is not clear. The main assumption about the existence of an optimal policy with a task-specific linear layer is not justified well.
* This paper claims that the proposed linear representation meta RL results in a better generalization to out-of-distribution tasks. However, it is not clear why the proposed method is better for generalization. I couldn't find any discussion about this observation or empirical study for clarifying why the proposed method is good for generalization.
* Faster runtime is one of the biggest strengths of the proposed method, but there is no discussion on why faster runtime matters for meta RL from the text.
* Experiments do not contain ablation studies for the proposed method. For example, I would curious about "predicting final linear layer vs predicting embedding" or "predicting from a set of (s, a, r, s') transitions vs predicting from a single (s, a, r, s') transitions".

Comments / Questions to authors
* I see a strong connection between this approach and PEARL, which predicts a low dimensional embedding from a set of transition data policy. What is the main benefit of predicting the last linear layer compared to predicting the low dimensional embedding?
* Does the adapter network change the linear layer parameter in every time step during meta-testing? What's the effect of changing parameters in every time steps vs predicting a single parameter from a good data point?
* In Figure 1, the proposed method showed about `-200` reward in Cheetah-Vel (hard) tasks. Is this a reasonably high reward to claim "generalization" to this task?

Recommendation
I recommend rejecting this paper because it is not clear why "linear representation meta RL" is better in general. I believe this is because the paper lacks a thorough discussion or empirical studies to clarify this question. Even though the proposed method showed slightly better performance on a few meta-RL benchmarks, it is not clear where the benefits come from and it is not clear whether we can expect a similar gain in more challenging problems.

### After the author feedback
I appreciate the authors for more control experiments and additional discussion in the manuscript. In the current manuscript, it is clearer that the "adapter network" predicting the "final linear layer" of the network is a unique component in this paper. I found that using an adapter network trained to predict linear layer parameters during training and used to infer parameters during testing is a setting that was not explored. However, I still see strong connections to previous works and I'm still not sure how to place this paper among such related works.

I see the approach in this paper looks similar to RL^2 approach e.g. meta-training an LSTM based policy and meta-testing on unseen tasks. In this case, inferring the hidden state of LSTM looks similar to what the adapter network does in this paper.  Two differences of this work from LSTM based RL^2 are 1) neural network architecture, and 2) training objectives.

One can construct a neural network architecture that is identical to an adapter network during testing time. One can train this network end-to-end during meta-training and use the exact same inference as this paper during meta-testing. This approach could be called RL^2 with a special neural network architecture that predicts the last layer of a neural network. If predicting the last layer of a neural network is an important component, it should be studied as an instance among variants of RL^2 with slightly different architecture. One practical concern about the architecture studied in this paper that this network may not scale to a case where the action space is large and the policy uses a large penultimate hidden state. In this case, predicting the parameter of a linear layer becomes very expensive. Because of the existence of this special case, I am not fully convinced about the claims that this method may work well in general.

This paper trains the adapter network to predict parameters of a neural network instead of training the adapter network end-to-end during meta-training. Because of this difference, the method in this paper cannot be called RL^2 and it could be claimed that this paper explores a method that is not explored previously. If the training method is an important component, there should be at least one ablation for this detail.

I still have a concern that it is not clear what's the main finding in this paper. Specifically, whether architecture is important or objective is important. I see that the paper already compared with RL^2, so adding an ablation study on using linear parameter prediction for RL^2 and discussing the relation to RL^2 would further improve the paper.

---

> ### Author Response · Authors · 2020-11-17
> **Response to AnonReviewer3**
>
> We thank AnonReviewer3 for the valuable comments and suggestions. Please find our response below (Broken into two responses to answer all questions).
>
> 1. The main assumption about the existence of an optimal policy with a task-specific linear layer is not justified well.
>
> We have added discussion on the assumption in the paper. Indeed, the linear policy spaces assumption have been adopted widely in literature and see good theoretical guarantee and empirical results, e.g. in paper "On the Sample Complexity of Reinforcement Learning with Policy Space Generalization" and references therein. Here we make it as an assumption for all the policies considered.
>
> Furthermore, we believe our strong experimental results among all the tasks including the new navigation task also justifies the validity of the assumption. We hope our algorithm helps provide direction for theoretical analysis and that the assumption is used to help provide intuition for the algorithm and explain potential results.
>
> 2. It is not clear why the proposed method is better for generalization. I couldn't find any discussion about this observation or empirical study for clarifying why the proposed method is good for generalization.
>
> Thanks for your question! We have added ablation studies and thorough discussions in Section 4.4 and Appendix C.1 to address this issue. We compare training and testing with adapter net vs without adapter net. It turns out that the two algorithms converge to a similar average return. However, the algorithm with adapter net requires much smaller sample complexity during adaptation and faster adaptation. This decomposes the role of our linear output layer structure and adapter net: since both algorithms converge to the same average return, we know that the linear output layers are the main reason for generalization, while adapter net is mainly for speeding up the algorithm and reducing sample complexity.
>
> We have added detailed discussion on the validity of FLAP in Section 4.4.3 and analyzed the reason for its small sample complexity, fast adaptation and good generalization. To summarize briefly from our revisions, our ablation studies show that (a) the small complexity is from the introduction of adapter net. We conjecture that during training, the adapter net learns the structure of optimal policies shared by all the training tasks;  and during adaptation, the predicted weights encourage the most efficient exploration over potential structure of optimal policies such that the average return converges in a fast and stable way; (b) the good generalization comes from the shared linear layer structure.  In traditional gradient-based meta-RL approaches like MAML, the new policies are generated from taking one gradient step from the original weights in training, which makes the new policies very similar to the old ones. In contrast, the new policy in FLAP can have a completely different last layer, which guarantees more flexibility in policies and thus better generalization.
>
>
> 3. There is no discussion on why faster runtime matters for meta RL from the text.
>
> Thank you for this point! We have added reasoning for why faster runtime matters in the introduction. In real-world applications like robotics, we would like to apply meta-RL in the hope that the robot can adapt to new environments and tasks as fast as possible. This not only requires a small sample complexity during adaptation, which corresponds to the number of interactions conducted with the environment, but also requires the robot to make an instant reaction given the feedback from the environment, which corresponds to the adaptation speed we considered here. We believe this is an important evaluation factor of meta-RL algorithms that have long been overlooked.
>
>
> 4. Experiments do not contain ablation studies for the proposed method.
>
> Thank you for this feedback! In Appendix C.1, we have added some important ablation studies that provide further analysis of our method on (a) Training and testing with adapter net vs without adapter net; (b) Training and testing with a single SARS tuple as adapter input vs a sequence of SARS as adapter input; (c) Training and testing with a single shared layer vs. shared two layers; (d) SAS input VS SARS input. The details are introduced in our general response to all reviewers as a parallel comment.

---

> > ### Author Response · Authors · 2020-11-17
> > **Response to AnonReviewer3 (Continued)**
> >
> > 5. I see a strong connection between this approach and PEARL. What is the main benefit of predicting the last linear layer compared to predicting the low dimensional embedding?
> >
> > First, our method is different from PEARL and all the prior meta-RL methods in that we do prediction during adaptation while all prior methods continue to “learn” via an objective during adaptation. By predicting the last layer weights directly, this enables us to do prediction during adaptation. To our knowledge, this is the first work on analyzing prediction on test tasks as opposed to solving a new optimization problem during adaptation. From our ablation study, we know that prediction helps improve the sample complexity during adaptation without sacrifice to performance. In combination with the expressive power of an entire layer in the neural network, we can see that learning the last linear layer is empirically better than learning the embeddings in comparison to these past meta-RL algorithms such as PEARL.
> >
> > Now, the benefits for predicting the last linear layer are as follows. By only predicting the linear set of weights, our method only needs 1 trajectory of data and is only limited in runtime by the feed-forward adapter network.  Typically past algorithms that predicted a low dimensional embedding required many samples to acquire a new policy and they also run a training process which requires the algorithm to run much longer compared to performing simple prediction to acquire the new policy. From the way our algorithm utilizes the adapter network, and further showcased in the ablation studies, we only need a single point really to obtain a set of reasonable weights to acquire a strong policy.
> >
> > This helps increase the competitiveness of our method not only in performance but also in how many samples from the new task it needs and how fast its adaptation process is. This indicates that FLAP may scale much better to more complex real-world problems compared to prior methods. Ultimately, FLAP helps shed new light for the meta-RL community that has previously focussed on utilizing learned information from training to continue the learning process during testing. (The latter of course shows massive limits in scalability especially in the face of fast-changing highly complex real-world problems)
> >
> > 6. Does the adapter network change the linear layer parameter in every time step during meta-testing? What's the effect of changing parameters in every time steps vs predicting a single parameter from a good data point?
> >
> > The network only changes the last linear layer weights for a fixed number of time steps and then only uses the latest set of weights for the rest of the episode.
> >
> > Our ultimate goal is to predict a single parameter from a good data point (or a sequence of data points). The changing in parameters during adaptation indeed helps us explore more structures and SARS tuples to find a good target data point in a faster way.
> >
> > 7. Is this -200 reward a reasonably high reward to claim "generalization" to this task?
> >
> > For the Half-Cheetah Velocity (Hard) task, -200 is a very high reward where the training reward is only slightly better. Our algorithm completely outperforms all prior methods in this problem and we claim then it strongly improves generalization (true generalization is relative of course to prior methods as there is no agreed upon “generalization” value).
> >
> > 8.  The paper lacks a thorough discussion or empirical studies to clarify why "linear representation meta RL" is better in general. It is not clear where the benefits come from and it is not clear whether we can expect a similar gain in more challenging problems.
> >
> > We hope that our answer for 1 & 2 (the new ablation studies in Appendix C.1 and discussions in Section 4.4) have addressed most of the concerns here. To summarize briefly, linear representation meta-RL is good in (a) small sample complexity; (b) good generalization; (c) fast adaptation speeds. Our ablation study shows that small sample complexity and fast run-time come from our use of an adapter network; while the generalization ability comes from linear representation learning. We have discussed the detailed reasons for them in Section 4.4.
> >
> > We also test out our results in a more challenging problem of navigation tasks and achieve great performance even though the SARS tuples are mostly similar and the rewards are sparse.

---

### Official Review · AnonReviewer2 · 2020-10-28
**The idea of weight prediction is not novel, and the generalization ability to out-of-distribution tasks requires further explanation and theoretical analysis.**

**Rating:** 6
**Confidence:** 4

**Review:**

## Summary:
This paper proposes a novel meta-RL algorithm called Fast Linearized Adaptive Policy (FLAP), which can adapt to both in-distribution and out-of-distribution tasks. FLAP is based on a strong assumption that some policies can be formalized as a linear combination of common task features. Based on this assumption, the authors design two modules. The first module is the policy networks with shared layers, which reduce the number of parameters to be optimized during meta testing. The other module is the Adapter Network, which generates task-specific policy weights from the sampled transition tuples. Experiments show that FLAP can handle certain tasks better than existing methods in terms of speed.

## Pros:
1. Adaptation speed. During meta-testing, FLAP uses prediction rather than optimization, which is an innovative step that takes advantage of NN's ability to infer fast in the meta-RL setting. If proved effective, this idea is worth further researches as it can accelerate the adaptation process and boost the performance on OOD tasks.
2. Performance of FLAP. The performance of FLAP is comparable to some SOTA algorithms on in-distribution tasks both in terms of memory usage and performance. When evaluating FLAP on OOD tasks, FLAP achieves better scores and stability in most task settings.

## Cons:
1. The validity of linear output layers. The key point that distinguishes FLAP from other meta-RL algorithms is the prediction of the policy weights. The author needs more theoretical analysis to prove the validity of this method on OOD tasks.
2. Generalization ability of the Adapter Network. Since the adapter only takes one transition tuple into consideration at each step, the resulting policy highly relies on the chosen transition tuple, which can cause high variance. Some tasks with sparse reward, e.g. navigation tasks, share a lot of similarities and even share the same dynamics. In such cases, the adapter can be confused and the performance of FLAP may deteriorate.

## Questions:
1. This paper makes an assumption that the Adapter Network can converge on the train tasks. It appears to me that the training process is unstable in some experiments, which may stop the adapter network from convergence. Would you provide more information about the training loss?
2. How does FLAP perform in sparse reward environments, e.g. navigations?
3. Predicting the weights of neural networks seems to be more difficult than predicting some more intuitive information, e.g. predicting rewards and states in MIER. How does the Adapter Network generalize to the OOD tasks even if it has not seen the states?

## Correction:
1. Both $\mathcal{D}$ and $\mathcal{T}$ refer to a set of tasks in different parts of the paper.
2. In Fig. 3: The definition of $x_{2}$ and $y_{2}$ should be explained.
3. In Paragraph 2, Section 2: For $\phi^k$ and $\pi^l$, k and l are the number of trajectories, which are inappropriate to denote $\phi$ and $\pi$, considering replacing it with the denotation of the trajectories.
4. In section “Experiments”: MIER-wR and the definition of the OOD tasks should be explained in the main body.
5. In section “Citation”: Dual citation of “Transfer learning for reinforcement learning domains”.
6. In appendix C.3: “PEARl” -> “PEARL”.

## Response to the author feedback:
We appreciate the authors for the extra effort to demonstrate the abiltiy of FLAP by adding more experimental results and discussions. Meta-reinforcement learning for adaptation to OOD tasks is an interesting field. Although the idea of FLAP is simple, the strong experimental results have demonstrated that by predicting weights rather than optimizing, FLAP is a fast and effective meta-RL algorithm for adaptation to OOD tasks compared to previous approaches that did not focus on this area. I believe FLAP will help draw more attention to this field and provide a direction for more theoretical analysis, thus I have increased the score.

---

> ### Author Response · Authors · 2020-11-16
> **Response to AnonReviewer2**
>
> We thank AnonReviewer2 for the valuable comments and suggestions. Please find our response below.
>
> 1. The author needs more theoretical analysis.
>
> Thank you for this suggestion. We have added in increased analysis and modified accordingly.
>
> First, we added new ablation studies on the adaptor network. We compare training and testing with adapter net vs without adapter net. It turns out that the two algorithms converge to a similar average return. However, the algorithm with adapter net requires much smaller sample complexity during adaptation and also faster adaptation. This decomposes the role of our linear output layer structure and adapter net: since both algorithms converge to the same average return, we know that the linear output layers are the main reason for generalization, while adapter net is mainly for speeding up the algorithm and reducing sample complexity.
>
> Second, we tried changing the input of the adapter from SARS tuple to SAS tuple (leave out the reward). It turns out that the adapter performs poorly with only SAS' tuples, which means that it does combine all the information of the SARS tuple to predict the weights.
>
> Third, we have added detailed discussion on the validity of FLAP in Section 4.4.3 and analyzed the reason for its small sample complexity, fast adaptation and good generalization. Our ablation studies show that (a) the small complexity is from the introduction of adapter net. We conjecture that during training, the adapter net learns the structure of optimal policies shared by all the training tasks;  and during adaptation, the predicted weights encourage the most efficient exploration over potential structures of optimal policies such that the average return converges in a fast and stable way; (b) the good generalization comes from the shared linear layer structure. In traditional gradient-based meta-RL approaches like MAML, the new policies are generated from taking one gradient step from the original weights in training, which makes the new policies very similar to the old ones. In contrast, the new policy in FLAP can have a completely different last layer, which guarantees more flexibility.
>
> We believe our strong experimental results on all the real-world tasks and the new navigation task have shown the validity of our linear layer assumption and the validity of this method on OOD tasks. Theoretical analysis may involve complicated and unrealistic assumptions that are indeed far from practice. This paper is also partially inspired by the theoretical analysis on learning in linear policy spaces in e.g. paper "On the Sample Complexity of Reinforcement Learning with Policy Space Generalization", where profound theory has been developed. We hope our algorithm helps provide direction for more theoretical analysis along this line.
>
>
> 2. How does FLAP perform in sparse reward environments?
>
> We conducted experiments on navigation tasks with sparse reward in Appendix C.2. It turns out that the performance of FLAP may deteriorate a little since the resulting policy highly relies on the chosen single transition tuple, which may lead to higher variance. However, it is still comparable to PEARL if we let the adapter net take a sequence of SARS tuples as input. This is due to that a sequence of SARS tuples provides more information on the contexts for adapter, making the resulting policy less dependent on the single chosen transition tuple.
>
> Based on this observation, we conduct more ablation studies on training and testing with a single SARS tuple as adapter input vs a sequence of SARS as adapter input. It turns out that both cases converge to a similar average return for all other tasks we considered in the paper. However, a longer sequence as input would reduce the variance, stabilize the adaptation at the sacrifice of adaptation speed. We have thoroughly discussed this phenomenon in the updated paper.
>
> 3. Would you provide more information about the loss for adapter?
>
> We have provided more information on the training loss for the adapter network in Appendix C.3. The training loss for adapter net converges in a stable way, which validates that our assumption on shared layers among policies holds in these instances.
>
>
> 4. How does the Adapter Network generalize?
>
> This case is indeed the example of Cheetah-Vel (hard) task, where the test tasks are very different from training. Our performance is still comparable to the state-of-the-art meta-RL algorithms.
>
> If only a small fraction of samples are unseen, we can take a sequence of tuples instead and average out their effect, like what we have done in the navigation task. On the other hand, we believe this is a big open problem for all meta-RL and transfer learning problems when most of the samples have unseen states that are completely different from the training stage. Also, since most of the environments here have continuous state and action spaces, the same state may not be seen twice, although they may be similar in some cases.

---

### Official Review · AnonReviewer1 · 2020-10-29
**Simple idea, compelling empirical results.**

**Rating:** 7
**Confidence:** 4

**Review:**

Meta-learning in RL, which is a form of transfer learning where we wish to rapidly transfer from experience across a set of related training tasks to novel tasks potentially from a related, but distinct distribution. In this work this is restricted to tasks which share the same state and action space. One application of such a problem is system identification in robotics, each physical robot may vary due to variations in manufacturing and wear, but we may require policies that can rapidly adapt to these variations.

The solution in this work is conceptually simple. We assume there exists some space such that policies for all tasks are linear in this space. A shared embedding is learned (across all training tasks) and all task specific policies are described by a task-specific weight and bias $w_i, b_i$ for task $i$ (eq 3). These are all learned using Soft Actor Critic.

Then there is the question of how we generalize to unseen test tasks. One possibility would be to attempt to learn the weights and bias $w_t, b_t$ for the test task rapidly through experience on the test task. However, the other idea of this work is to use an ``adapter network'' to predict, from a tuple of transition $(s, a, r, s')$ the corresponding $w_t, b_t$ for this task. This network is trained using regression on the training tasks (where the true values of $w_i, b_i$ are learned using SAC). Then, given potentially even a single transition on a novel task this network can output an estimate of $w_t, b_t$ thus defining a policy for the test task.

## Overall

The approach in this work is relatively straightforward and many variations in using embedding spaces for policies have been previously considered. This is not a criticism, revisiting and modifying existing ideas and demonstrating that they are able to obtain state of the art performance is useful. This work demonstrates empirical performance on a set of meta-learning tasks that is competitive. The main suggested improvements are further discussion for related work, clarifications and experiments and/or discussions of limitations of this approach.

## Feedback

The RL algorithm used to for training is Soft Actor Critic which approximately optimizes the maximum entropy RL objective. It would be helpful to discuss whether this is important for this approach. For example, how well does this method work if deterministic policies are used instead. Is the entropy term crucial for learning generalizable embeddings? [e.g. see 4,5 for arguments about why max ent framework can be useful for generalization].

The adapter network potentially makes a new prediction every timestep. How is this information averaged over timesteps, is always the latest prediction used?

It would be helpful to discuss (and ideally show experiments) where this approach performs poorly. For example, it would appear that the adapter network should not work on sparse rewards tasks (where the transition function is unchanged between tasks) or other situations where most experience tuples do not provide information regarding the specific task.

There are many pieces of related work using shared embeddings to transfer between tasks that should be discussed. These include UVFAs [1] (representations for linear value functions), successor features (learning representations such that rewards are linear in this space) [2], non-linear embedding spaces for tasks allowing generalizing to new state and action spaces [3] among others. [4] section 4.4/4.5 provides an overview of many approaches to learning representations the are reusable between tasks

## Minor issues
- Mendonca et al. seems to be listed twice in the reference list.

## References
[1] Schaul, T., Horgan, D., Gregor, K., & Silver, D. (2015, June). Universal value function approximators. In International conference on machine learning (pp. 1312-1320).

[2] Barreto, André, et al. "Successor features for transfer in reinforcement learning." Advances in neural information processing systems. 2017.

[3] Gupta, Abhishek, et al. "Learning invariant feature spaces to transfer skills with reinforcement learning." arXiv preprint arXiv:1703.02949 (2017).

[4] Hunt, Jonathan, et al. "Composing entropic policies using divergence correction." International Conference on Machine Learning. PMLR, 2019.

[5] Haarnoja, Tuomas, et al. "Composable deep reinforcement learning for robotic manipulation." 2018 IEEE International Conference on Robotics and Automation (ICRA). IEEE, 2018.

---

> ### Author Response · Authors · 2020-11-16
> **Response to AnonReviewer1**
>
> We thank AnonReviewer1 for the detailed comments and suggestions. Please find our response below.
>
> 1. How well does this method work if deterministic policies are used instead? Is the entropy term crucial for learning generalizable embeddings?
>
> We build upon the SAC algorithm given its state-of-the-art performance which helps give FLAP such a competitive sample complexity. It is also used in other meta-RL algorithms like PEARL for good sample efficiency and stability. However, any algorithm that utilizes a neural network representation of the policy can be substituted for the SAC algorithm so the entropy term is not absolutely crucial.
>
> To address the question on deterministic policies, we note that we baseline on continuous control Mujoco problems. It is much more typical to utilize stochastic policies on these highly complex environments as composed to deterministic policies to have an improvement in performance. This is also the common practice for other meta-RL algorithms and benchmarks.
>
> 2. (And 3) How is this information averaged over timesteps, is always the latest prediction used? It would be helpful to discuss (and ideally show experiments) where this approach performs poorly on e.g. navigation tasks.
>
> Thank you for the questions. Our new experiments are indeed inspired by this question and other reviewers' comments!
>
> In our new ablation study, we consider two different designs of adapter net, one takes only the most recent SARS tuple and uses the latest prediction as the final weight, the other takes a sequence of the most recent SARS tuple and uses the latest prediction as the final weight. The second design actually aims at averaging over the recent timesteps by taking all of them as input of the adapter net. It turns out that both cases converge to a similar average return. However, a longer sequence as input would reduce the variance, stabilize the adaptation at the sacrifice of adaptation speed. Since for almost all the tasks we considered, the original FLAP algorithm works well with small variance, we don't need to sacrifice the speed in exchange of a smaller variance.
>
> However, the story is different when it comes to the case of navigation tasks. We conducted experiments on navigation tasks with sparse rewards in Appendix C.2. It turns out that the performance of the original FLAP algorithm deteriorates a little since the resulting policy highly relies on the chosen single transition tuple, which may lead to higher variance. However, it is still comparable to PEARL if we let the adapter net take a sequence of SARS tuples as input. This is due to the fact that a sequence of SARS tuples provides more information on the contexts for adapter, making the resulting policy less dependent on the single chosen transition tuple.
>
> 4. There are many pieces of related work using shared embeddings to transfer between tasks that should be discussed.
>
> We have added these in the related work. Thank you for the suggestion!

---

### Author Response · Authors · 2020-11-17
**Paper Revision**

We thank all the reviewers for the helpful and insightful comments and suggestions, which helps us significantly improve the paper in terms of experiments and analysis.  We have conducted a branch of new experiments accordingly to justify our FLAP algorithm. The main changes are summarized as follows, ordered in importance:

1. We added ablation studies in Appendix C.1. In short, we compare the following scenarios:

(a). Training and testing with adapter net vs without adapter net. It turns out that the two algorithms converge to a similar average return. However, the algorithm with adapter net requires much smaller sample complexity and faster adaptation.

(b). Training and testing with a single SARS tuple as adapter input vs a sequence of SARS as adapter input. It turns out that both cases converge to a similar average return. However, a longer sequence as input would reduce the variance, stabilize the adaptation at the sacrifice of adaptation speed.

(c). Training and testing with a single shared layer vs. two shared layers. It turns out shared two layers may not converge to the global minimum due to the additional degree of freedom.

(d). SAS’ input VS SARS input. It turns out that without reward information, the adapter network performs nearly randomly, which suggests that the adapter network is really learning meaningful information from SARS pairs.

2. We conducted experiments on navigation tasks with sparse reward in Appendix C.2. It turns out that the performance of FLAP may deteriorate a little since the resulting policy highly relies on the chosen single transition tuple, which may lead to higher variance. However, it is still comparable to PEARL if we let the adapter net take a sequence of SARS tuples as input. This is due to the fact that a sequence of SARS tuples provides more information on the contexts for adapter, making the resulting policy less dependent on the single chosen transition tuple.

3. We discuss all the new experiments mentioned above in Section 4.4.1 and 4.4.2. We also explain in detail why FLAP and adapter have superior performance based on our new ablation studies in Section 4.4.3.

4. We include the training loss for adapter net in Appendix C.3. It converges in a fast and stable way.

5. In the introduction, we added in justification for why runtime is important for meta-RL algorithms, and added in more references and discussion on related work in learning embeddings.

6. We also have fixed all typos and cleaned up the notation.

---

### Decision · Program_Chairs · 2021-01-07
**Final Decision**

**Decision:**

Reject

**Comment:**

Summary of discussions: R1 was positive on the paper in their initial evaluation, and although dissatisfied with the author's feedback, continued to support the paper. I agree with R1's assessment that other reviewers' call for more theory is somewhat unfair, considering the fact that very similar papers don't usually include theoretical justification beyond intuitive motivation.

By contrast, R3 is the most negative on the paper, leaning towards rejection. The main concern is that open questions remain as to whether the reported performance can be attributed to the architecture, or the loss function proposed. This is an important point to clarify, and further ablation studies would make the paper stronger.

After considering the strengths and weaknesses of this work, the final decision was to reject. Authors are encouraged to improve this promising work and resubmit to a future venue.